# Retinex-Based Fast Algorithm for Low-Light Image Enhancement

**DOI:** 10.3390/e23060746

**Published:** 2021-06-13

**Authors:** Shouxin Liu, Wei Long, Lei He, Yanyan Li, Wei Ding

**Affiliations:** School of Mechanical Engineering, Sichuan University, Chengdu 610065, China; liushouxin@stu.scu.edu.cn (S.L.); long_wei@scu.edu.cn (W.L.); markushe_scu@163.com (L.H.); dingwei1995@stu.scu.edu.cn (W.D.)

**Keywords:** Retinex, image enhancement, gamma correction, low-light image, HSV color space

## Abstract

We proposed the Retinex-based fast algorithm (RBFA) to achieve low-light image enhancement in this paper, which can restore information that is covered by low illuminance. The proposed algorithm consists of the following parts. Firstly, we convert the low-light image from the RGB (red, green, blue) color space to the HSV (hue, saturation, value) color space and use the linear function to stretch the original gray level dynamic range of the V component. Then, we estimate the illumination image via adaptive gamma correction and use the Retinex model to achieve the brightness enhancement. After that, we further stretch the gray level dynamic range to avoid low image contrast. Finally, we design another mapping function to achieve color saturation correction and convert the enhanced image from the HSV color space to the RGB color space after which we can obtain the clear image. The experimental results show that the enhanced images with the proposed method have better qualitative and quantitative evaluations and lower computational complexity than other state-of-the-art methods.

## 1. Introduction

Images captured with a camera in weakly illuminated environments are often degraded. For example, these types of images with low contrast and low light, reduce visibility. The object and detail information cannot be captured, which can reduce the performance of image-based analysis systems, such as computer vision systems, image processing systems and intelligent traffic analysis systems [1,2,3].

In order to address the above problems, a great number of low-light image enhancement methods have been proposed. Generally, the existing methods can be divided into three categories, namely the HE-based (histogram equalization) algorithm, Retinex-based algorithm and non-linear transformation [4,5,6]. The HE-based algorithm is the simplest method; the main idea of this method is to adjust illuminance by equalizing the histogram of the input low-light image. To address the shortage of conventional HE algorithms, over enhancement and loss of detail information, a great number of improved and HE-based methods have been proposed, such as contrast-limited equalization (CLAHE), bi-histogram equalization with a plateau limit (BHE), exposure-based sub-image histogram equalization (ESIHE) and exposure-based multi-histogram equalization contrast enhancement for non-uniform illumination images (EMHE) [7,8,9,10,11,12]. However, HE-based methods neglect the noise hidden in the dark region of low-light images. The Retinex model is a color perception model of human vision, which consists of illumination and reflectance [13,14]. The aim of Retinex-based algorithms is to estimate the right illumination image or reflectance image from its degraded image by different filters to achieve low brightness enhancement [15,16]. Some classic algorithms are single-scale Retinex (SSR) and multi-scale Retinex (MSR). In order to solve color distortion, multi-scale Retinex with color restoration (MSRCR) was proposed, which introduced color restoration in multi-scale Retinex. After that, some improved algorithms introduced different types of filters to replace the traditional Gaussian filter, such as the improved Gaussian filter, improved guided filter, bright-pass filter and so on [17,18,19]. Even though image texture details can be restored well via the Retinex-based method, the halo effect is introduced into enhanced images. Common non-linear functions are gamma correction, sigmoid transfer function and logarithmic transfer function [20,21,22]; these types of methods are pixel-wise operations for natural low-light images. Compared with other non-linear functions, the gamma transfer function is wildly used in the field of image processing, but the limitation of gamma correction is that if the parameter γ is too small, it will amplify the noise of the target image; by contrast, if the parameter γ is close to 1, satisfactory enhanced results will not be obtained. Therefore, estimating a suitable γ value is the key to obtaining satisfactory enhanced results.

In this paper, we utilize the gamma transfer function to estimate the illumination and achieve brightness enhancement via the Retinex model. The enhanced image achieves satisfactory light enhancement and global brightness equalization; thus, our method can restore more information than other methods. The final experimental results show that compared with other state-of-the-art methods, the enhanced images through our algorithm have better qualitative and quantitative evaluations. Some examples of natural low-light images and enhanced images with the proposed RBFA method are shown in Figure 1. All low-light images in Figure 1 were captured by the authors of this paper.

The rest of this paper is organized as follows: Section 2 describes the corresponding works of the proposed algorithm in this paper. In Section 3, the details of the proposed method are introduced. Section 4 presents the comparative experiment results with other state-of-the-art methods and describes the computational complexity comparison. The work is concluded in Section 5.

## 2. Related Work

We introduce the Retinex model, gamma correction and HSV color space in this section, which construct the basis of our method.

### 2.1. Retinex Model

The classical Retinex model assumes that the observed image consists of reflectance and illumination. The Retinex model can be expressed as follows [23].
(1)H=R • L 
where H is the observed image, R and L represent the reflectance and the illumination of the image, respectively. The operator ‘•’ denotes the multiplication. In this paper, we utilize the logarithmic transformation to reduce computational complexity. We can obtain the following expression.
(2)log(H)=log(R•L) 

Finally, we can obtain Equation (3) to estimate the reflectance in the HSV color space.
(3)log(R)=log(V)−log(L) 

### 2.2. Gamma Correction

The gamma transfer function is wildly used in the field of image processing, and the corresponding gamma transfer function can be expressed as follows [24,25].
(4)$$g\left(x,y\right)=u{\left(x,y\right)}^{\gamma }\;$$
where g(x,y) denotes the gray level of the enhanced image at pixel location (x,y), u(x,y) is the gray level of the input low-light image at pixel location (x,y), and γ represents the parameter of the gamma transfer function. The shape of the gamma transfer function can be affected by parameter γ; the influence of different values of γ is shown in Figure 2.

According to the Figure 2, we can see that the enhanced gray level increases monotonically with decreased parameter γ; if we want to achieve a higher value of the gray level, we have to let the size of parameter γ fall within the range from 0 to 1. Contrastingly, the enhanced gray level decreases monotonically with increased parameter γ.

### 2.3. HSV Color Space

The HSV color space consists of a hue component (H), saturation component (S) and value component (V) [26,27]. The value component represents the brightness intensity of the image. The advantage of the HSV color space is that any component can be adjusted without affecting each other [28]; more specifically, the input image is transferred from the RGB (red, green, blue) color space to the HSV color space, which can eliminate the strong color correlation of the image in the RGB color space. Therefore, this work is based on the HSV color space [29]. Commonly, image enhancement in RGB color space need to process R, G and B, three components, but we only need to process the V component in this work. Therefore, this will greatly reduce the image processing time.

## 3. Our Approach

The details of proposed algorithm are described in this section. Based on the descriptions in Section 2.2, in this work we only focus on the V component to adjust the brightness of the low-light image; the flowchart of the proposed method is shown in Figure 3. We choose an image named “Arno” to illustrate the enhancement process of the proposed method, the processing of image enhancement and corresponding histograms are shown in Figure 4.

In our method, we use gamma correction to estimate the illumination and the Retinex model to achieve brightness enhancement. Compared with using filters to estimate the illumination, using gamma correction to estimate the illumination can effectively reduce the computational time. The key to gamma correction is to compute the value of the gamma parameter; the details of the gamma parameter determined are described as follows.

### 3.1. Brightness Enhancement

The gray levels of a low-light image are mainly concentrated in the low gray level area, and the dynamic range of low gray levels is very narrow. Combing Figure 2, we can see that the higher the gray level dynamic range of the input image, the higher the gray level dynamic range of the output image. Therefore, we use linear enhancement to stretch the gray level dynamic range before gamma correction, and we make the value of the stretched gray level fall within the range of (0, 1) to prevent over-enhancement. The used linear function in this paper can be expressed as follows.
(5)Vmax=max(V(x,y)) 
(6)$$V1\left(x,y\right)=\frac{1}{Vmax}*V\left(x,y\right)\;$$
where Vmax denotes the maximum pixel value of V component, max(.) denotes take the maximum value of V(x,y), V(x,y) is the pixel value of the original V component at location (x,y), V1(x,y) is the enhanced pixel value at location (x,y) and ‘∗’ represents the multiplication.

The maximum value of the low-light image is usually lower than 1; we can infer that $\frac{1}{Vmax}>1$, so this linear function can stretch the dynamic range of the low-light image, and we also can obtain that V1(x,y)≤1.

After the gray level dynamic range is stretched, we adopt gamma correction to estimate illumination. For a low-light image, the lower the brightness intensity, the lower the gray level. Therefore, we take this feature into consideration. First, based on the global histogram, we compute the mean gray level value, which can reflect the overall brightness level to a certain extent. The corresponding computational formula is expressed as Equation (7), and we can obtain the mean gray level value via this equation.
(7)$$m=\frac{\sum_{i=0}^{L}P\left(i\right)*i}{\sum_{i=0}^{L}P\left(i\right)}\;$$
where m is the mean value of gray levels, L denotes the maximum value of gray levels of an image and P(i) is the histogram of gray level i.

In this paper, we assume that the gray levels more than zero and less than m+1 are the extreme low gray levels. In fact, this part of the gray level is the key to determine the mean gray level of the low-light image. Based on the above descriptions, we design a formula to convert the gray level of this part into a constant, and use this constant to compute the gamma value. The corresponding transfer formula is expressed as Equation (8).
(8)$$c=\frac{\sum_{i=1}^{m}P\left(i\right)*i}{128*\sum_{i=1}^{m}P\left(i\right)}\;$$
where *c* is the value of conversion result and *c* is a positive number. Low-light images may have similar mean values, which will lead to similar *c* values. In order to enlarge the difference of *c* values among different images, we use the following expression to enlarge *c* values.
(9)$$c1=\frac{1}{1+e^{-c}}\;$$
where c1 represents the enlarged *c* value. In addition, we also think that the focus of brightness enhancement lies in the low gray level area rather than the high gray level area. Therefore, we take the distribution of the low gray level as one of the important bases for estimating the gamma value. In order to calculate the distribution of the low gray level, the cumulative distribution function (CDF) is used to calculate the distribution of the gray level in this part. In this paper, we consider the gray level less than 128 to be the low gray level area.
(10)$$cdf\left(j\right)=\overset{j}{\underset{0}{\sum }}pdf\left(i\right)\;$$
(11)$$pdf\left(i\right)=\frac{p\left(i\right)}{M*N}\;\left(11\right)$$
where p(i) is the number of pixels that have gray level i, M and N are the length and width of the image, j is the threshold point of CDF and we set j equals to 128. Then we weigh the CDF value with the c1 value to obtain the gamma parameter value.
(12)γ=w∗c1+(1−w)∗cdf 
where γ represents the gamma parameter, w is the weighted value and equals to 0.48. Combining Equations (4), (6) and (12), we can get the final expression as follows.
(13)$$VL\left(x,y\right)=V1{\left(x,y\right)}^{w*c1+\left(1-w\right)*cdf}\;$$
where VL(x,y) denotes the pixel location (x,y) of illumination image. Combing Equations (3) and (13), we can get the reflectance, and it is shown as follows.
(14)log(R)=log(V)−log(VL) 

We get the enhanced V component as follows:(15)VE=exp(log(V)−log(VL)) 

The enhanced V component and corresponding histogram are shown in Figure 4c.

### 3.2. Dynamic Range Expansion

After brightness enhancement, the pixel values are easily concentrated in the higher gray level range, which leads to the grayscale dynamic range becoming narrow with low contrast in the enhanced image. We can adjust the contrast of the image by enlarging the V component gray level [30,31]. In order to avoid pixels values concentrated in the higher gray level range, we use a piecewise function to further stretch the gray level dynamic range to achieve dynamic range expansion. The corresponding expression can be expressed as follows.
(16)$$VE^{\prime }\left(x,y\right)=\{\begin{array}{cc} VE\left(x,y\right), & \;VE\left(x,y\right)\geq 0.5\\ 2*{\left(VE\left(x,y\right)\right)}^{2}, & \;VE\left(x,y\right)<0.5 \end{array}\;$$
The dynamic range enlarged V component and corresponding histogram are shown in Figure 4d.

### 3.3. Saturation Adjustment

In addition to brightness, the color saturation also directly affects the visual experience. In the HSV color space, the mean value of the S component and V component of a clear image should be approximately equal [32,33]. However, with the adjustment of brightness, the mean value of the V component changes greatly, which affects the image color. Based on the mean difference between the V component and the S component, Formula (20) is designed to adjust the S component. The details of our method are described as follows. Firstly, we use Equation (17) to compute the mean difference between the V component and S component.
(17)$$VES=VE^{\prime }mean-Smean\;$$
where VES is the mean difference, $VE^{\prime }mean$ is the mean value of enhanced V component and Smean is the mean value of S component. The expression used to compute $VE^{\prime }mean$ is shown below.
(18)$$VE^{\prime }mean=\frac{\sum_{0}^{i}VE^{\prime }\left(i\right)*i}{M*N}\;$$
where i denotes the gray level, and $VE^{\prime }\left(i\right)$ is the number of pixels that have gray level i. M and N are the length and width of the image. Similiarly, we can get Equation (19) to compute the Smean.
(19)$$Smean=\frac{\sum_{0}^{i}S\left(i\right)*i}{M*N}\;$$
where i denotes the gray level, S(i) is the number of pixels that have gray level i. From the above description, we adjust the S component value to reduce the mean difference value between the *VE’* component and S component to achieve the purpose of color saturation adjustment. After VES is obtained, we use it to adjust the S component. According to Section 2.2, if we want to enlarge the value of the S component, we have to ensure that the gamma parameter lies in the range (0,1). On the contrary, we need to ensure that the parameter value is greater than 1 to reduce the value of the S component. Therefore, we use Equation (20) to achieve this step.
(20)$$S1\left(x,y\right)=S{\left(x,y\right)}^{1+{\left(-1\right)}^{2-n}*\left({\left|VES\right|}^{2}+\left|VES\right|\right)},\;n=\{\begin{array}{c} 0\;VES<0\\ 1\;VES\geq 0 \end{array}\;$$
where S1(x,y) denotes the pixel location (x,y) of the adjusted S component, and S(x,y) is the pixel location (x,y) of the original S component. According to Equation (17), we can see that if VES<0, we know that VEmean<Smean, so we need to reduce the value of the S component. Meanwhile, from Equation (20) we know that n=0 and $1+{\left(-1\right)}^{2-n}*\left({\left|VES\right|}^{2}+\left|VES\right|\right)>1$, then we get S1(x,y)<S(x,y). Similarly, we can see that when VES>0, we also can get S1(x,y)>S(x,y). The original S component and corresponding histogram are shown in Figure 4e and the adjusted S component and corresponding histogram are shown in Figure 4f.

## 4. Comparative Experiment and Discussion

This section describes the comparative experiment with the existing methods and experimental results. The comparative methods used include the LECARM algorithm [34], FFM algorithm [7], LIME algorithm [17], AFEM algorithm [1], JIEP algorithm [15] and SDD algorithm [35]. All comparative experiments are performed in MATLAB R2020b on a PC running Windows 10 with an Intel (R) Core (TM) i7-10875H CPU @ 2.30 GHz and 16 GB of RAM. Due to the length limitation of this paper, we use 10 images to illustrate the comparative results; the reference images are shown in Figure 5. All test images and reference images come from the public MEF dataset [36], which, in total, include 24 low-light images.

### 4.1. Computational Time Comparison

We test the time consumed for different algorithms to process different size images, and the test results are shown in Table 1.

In Table 1, the shortest times are highlighted in bold case values, and the second-shortest times are highlighted with underlined values. Table 1 shows that the proposed method takes the shortest time for processing each image due to the lowest computational complexity, in comparison to both the FFM method and SDD method, which consume the longer time. We also can learn that JIEP’s time consumption is higher than the AFEM method and less than the FFM method. The time consumptions of AFEM, LECARM and LIME are similar because of the same computational complexity. Generally, the proposed RBFA algorithm consumes the least time on average, and the processing speed of the image is the fastest.

We made the data in Table 1 into a line chart to analyze the computational comparison of different methods as shown in Figure 6**.** Figure 6 shows that the computational complexity of the proposed method RBFA is *O*(*N*), and it is the lowest among all the methods, in comparison to SDD’s computational complexity, which is the highest. The computational complexity of the SDD method is *O*(*N*^2^), which results in the SDD method costing more time on image processing. The computational complexity of both the FFM method and JIEP method are *O*(*N*log*N*), but the time increment of the FFM method is higher than the JIEP method. The computation complexity of AFEM, LECARM, LIME and the proposed method MFGC are *O*(*N*); although the computational complexity is the same, the time increment of the proposed algorithm is the smallest with the same amount of data, proving that the proposed method has the lowest computational complexity.

### 4.2. Visual Comparison

Although the results of the LECARM method preserve the original hue and saturation and have a higher brightness intensity of each image, this algorithm easily results in non-uniform global light, further decreasing the visual experience, such as in the mid area of Figure 7, Figure 8, Figure 9 and Figure 10, which have higher illumination than other areas. The SDD method results show that there are some regions that become blurred, such as in the middle area of Figure 10 and Figure 11. From the enhanced images, we can see that the performance of FFM is unstable, because some results of the FFM method are inadequate brightness enhancement, for instance, the whole area of Figure 12 and Figure 13. From the results of the JIEP method, we can see that this algorithm is focused on normal under exposure and does not perform well in extreme low illuminance regions, such as in the middle region of Figure 8 and the bottom region of Figure 14. It is clear that the results of the LIME method show uneven brightness and over-enhancement in some areas, such as the lower middle area of Figure 10, middle area of Figure 14 and Figure 15 and the wall in the Figure 16. The results of the AFEM algorithm are not satisfactory because the brightness increment is too small to restore the details covered with dark regions, such as the bottom area of Figure 14 and Figure 16. As we can see, the results of the proposed RBFA method achieved the global brightness balance after enhancement via the proposed method; the color retained is more natural than the other methods.

### 4.3. Objective Assessment

Because human eyes often lose some details when we observe a picture, we choose one no-reference image quality assessment metric (perception-based image quality evaluator (PIQE)), three full reference image quality measure metrics (mean-squared error (MSE), structural similarity (SSIM), and peak signal-to-noise ratio (PSNR)) and lightness order error (LOE) to measure the quality of the enhanced images. The results of the different quality measure methods are shown in Table 2; these values represent the average value. The best scores are highlighted in bold case values, and the second-best scores are highlighted with underline values.

We can see from Table 2 that the proposed method obtained the best score four times and second-best score once. As shown in Table 2, the PIQE values of different methods fall within the range from 38.601 to 51.457, which means that the quality of all enhanced images is very similar and close, and the enhanced images via the proposed method obtained the best score. The smaller the LOE value, the more natural the enhancement effect. We can see that the LOE value of the proposed method is the best. This also means that the naturalness of the preservation of the proposed method is efficient. MSE is calculated by taking the average of the square of the difference between the reference image and enhanced image; the smaller the value is, the higher the similarity between the reference image and the enhanced image. The result of the proposed method is only 4.59 lower than the best LIME result. SSIM assesses the visual impact of three characteristics of an image: luminance, contrast and structure. The bigger the SSIM value, the higher the image quality; we see that the enhanced image via the proposed method preserved the highest similarity to the reference image. We know that the PSNR value of the proposed method is also the highest, which means that our method is useful for low-light image enhancement. Generally, the image quality enhanced by the proposed method is better than other comparative methods.

## 5. Conclusions

We proposed the Retinex-based fast enhancement method in this paper. This method can address uneven brightness and greatly improve the brightness of low-light areas. The proposed method is more efficient. In general, the proposed RBFA algorithm performance is better than other state-of-the-art methods, combining the results of the comparative experiment, computational complexity comparison and quality assessment. In other words, the proposed RBFA method is a simple and efficient low-light image-enhancement algorithm.

## Figures and Tables

**Figure 1 entropy-23-00746-f001:**
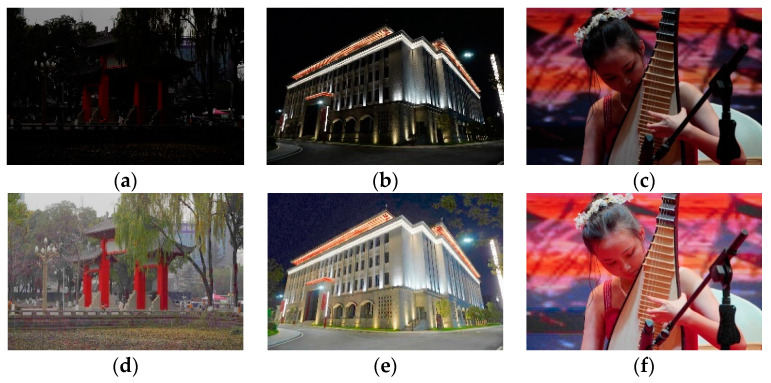
Top row (**a**–**c**): natural low-light images, bottom row (**d**–**f**): enhanced images with our proposed RBFA method.

**Figure 2 entropy-23-00746-f002:**
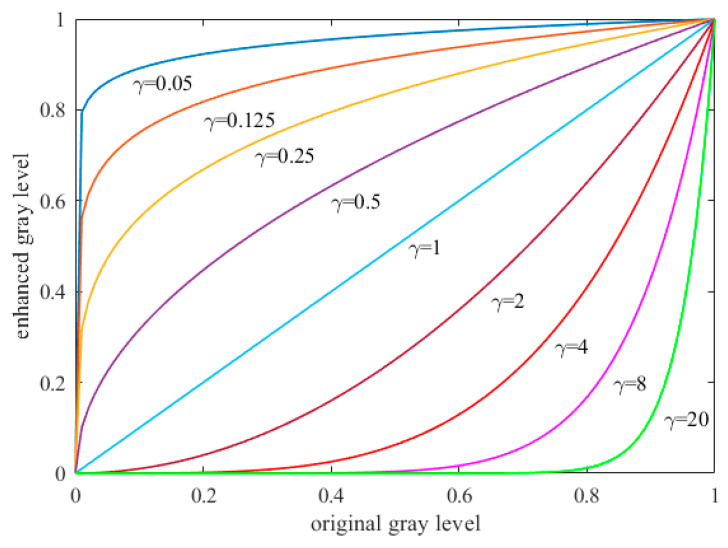
The shapes of gamma functions with different *γ* values.

**Figure 3 entropy-23-00746-f003:**
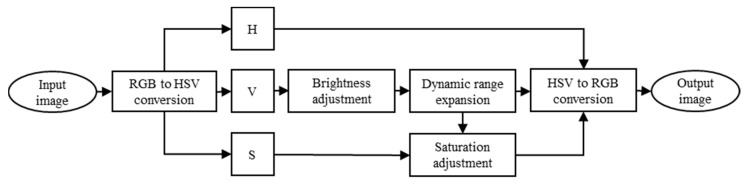
The flowchart of the proposed method.

**Figure 4 entropy-23-00746-f004:**
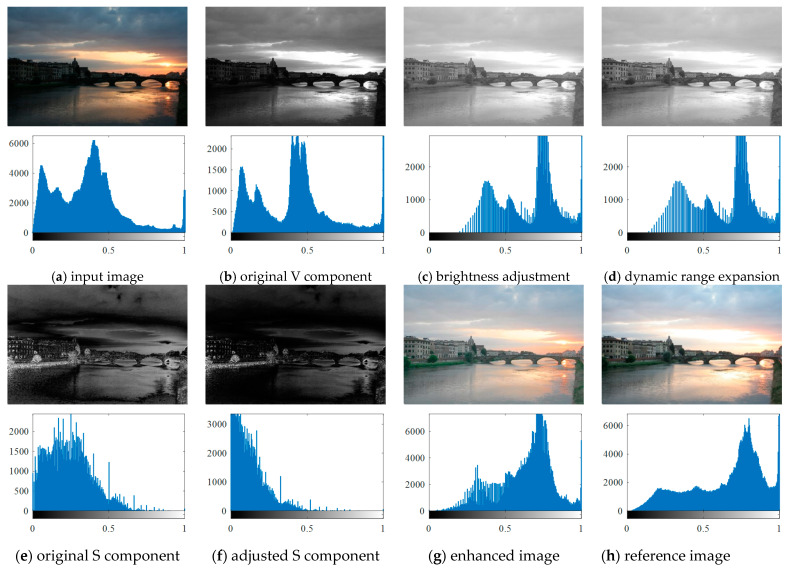
Low-light image enhancement process and corresponding grayscale histograms.

**Figure 5 entropy-23-00746-f005:**
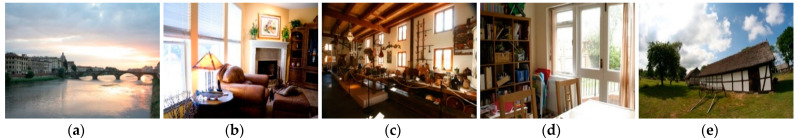
Reference images (**a**–**j**).

**Figure 6 entropy-23-00746-f006:**
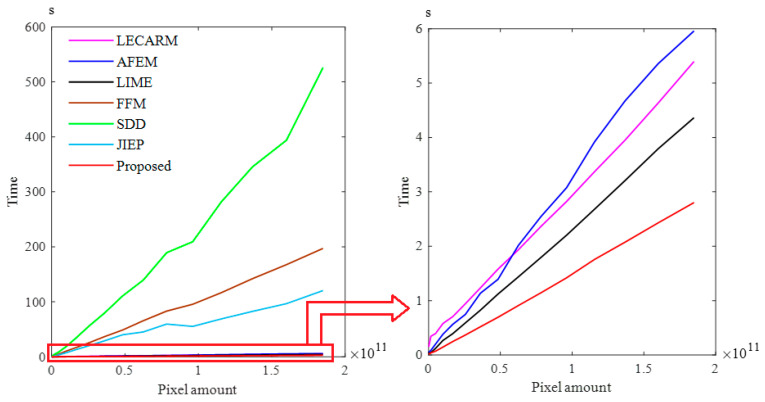
Result of computational complexity comparison.

**Figure 7 entropy-23-00746-f007:**
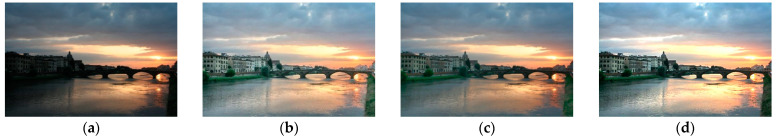
Comparing enhanced results of Arno with different methods. (**a**) Input image, (**b**) enhanced with LECARM, (**c**) enhanced with FFM, (**d**) enhanced with LIME, (**e**) enhanced with AFEM, (**f**) enhanced with JIEP, (**g**) enhanced with SDD, (**h**) enhanced with proposed RBFA method.

**Figure 8 entropy-23-00746-f008:**
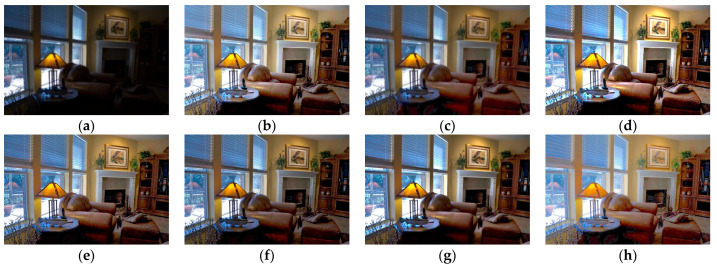
Comparing the enhanced results of Room with different methods. (**a**) Input image, (**b**) enhanced with LECARM, (**c**) enhanced with FFM, (**d**) enhanced with LIME, (**e**) enhanced with AFEM, (**f**) enhanced with JIEP, (**g**) enhanced with SDD, (**h**) enhanced with proposed RBFA method.

**Figure 9 entropy-23-00746-f009:**
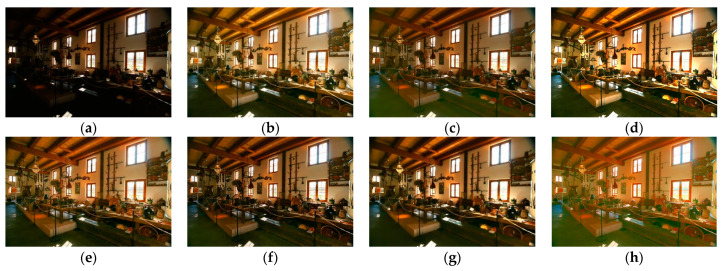
Comparing enhanced results of Farmhouse with different methods. (**a**) Input image, (**b**) enhanced with LECARM, (**c**) enhanced with FFM, (**d**) enhanced with LIME, (**e**) enhanced with AFEM, (**f**) enhanced with JIEP, (**g**) enhanced with SDD, (**h**) enhanced with proposed RBFA method.

**Figure 10 entropy-23-00746-f010:**
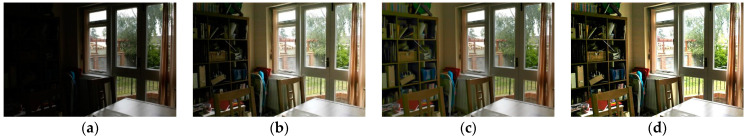
Comparing enhanced results of House with different methods. (**a**) Input image, (**b**) enhanced with LECARM, (**c**) enhanced with FFM, (**d**) enhanced with LIME, (**e**) enhanced with AFEM, (**f**) enhanced with JIEP, (**g**) enhanced with SDD, (**h**) enhanced with the proposed RBFA method.

**Figure 11 entropy-23-00746-f011:**
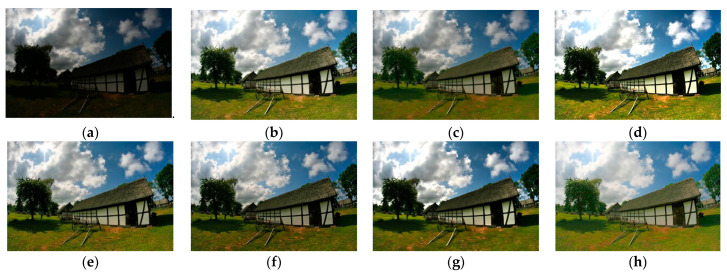
Comparing enhanced results of Cru with different methods. (**a**) Input image, (**b**) enhanced with LECARM, (**c**) enhanced with FFM, (**d**) enhanced with LIME, (**e**) enhanced with AFEM, (**f**) enhanced with JIEP, (**g**) enhanced with SDD, (**h**) enhanced with proposed RBFA method.

**Figure 12 entropy-23-00746-f012:**
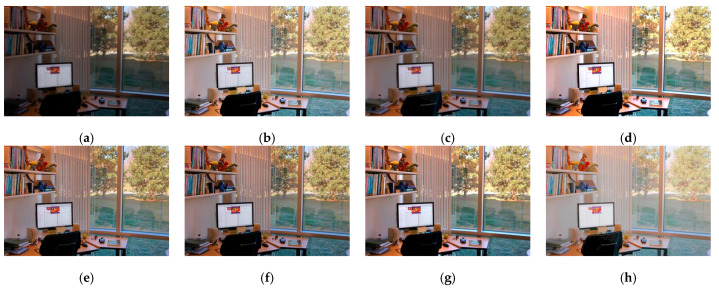
Comparing enhanced results of Office with different methods. (**a**) Input image, (**b**) enhanced with LECARM, (**c**) enhanced with FFM, (**d**) enhanced with LIME, (**e**) enhanced with AFEM, (**f**) enhanced with JIEP, (**g**) enhanced with SDD, (**h**) enhanced with proposed RBFA method.

**Figure 13 entropy-23-00746-f013:**
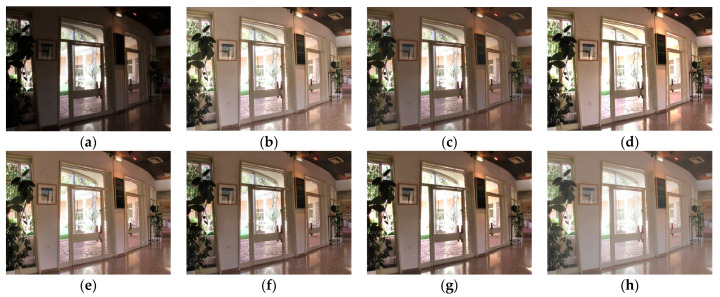
Comparing enhanced results of Door with different methods. (**a**) Input image, (**b**) enhanced with LECARM, (**c**) enhanced with FFM, (**d**) enhanced with LIME, (**e**) enhanced with AFEM, (**f**) enhanced with JIEP, (**g**) enhanced with SDD, (**h**) enhanced with proposed RBFA method.

**Figure 14 entropy-23-00746-f014:**
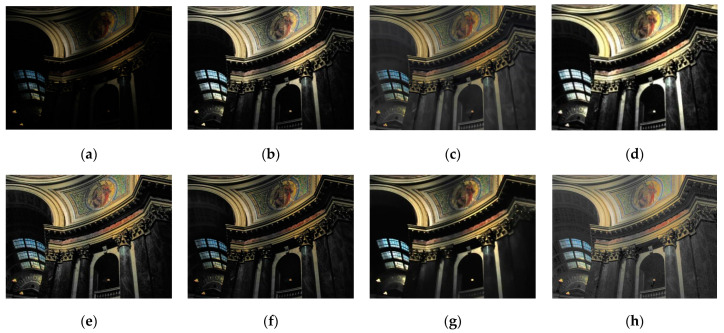
Comparing enhanced results of Capitol with different methods. (**a**) Input image, (**b**) enhanced with LECARM, (**c**) enhanced with FFM, (**d**) enhanced with LIME, (**e**) enhanced with AFEM, (**f**) enhanced with JIEP, (**g**) enhanced with SDD, (**h**) enhanced with proposed RBFA method.

**Figure 15 entropy-23-00746-f015:**
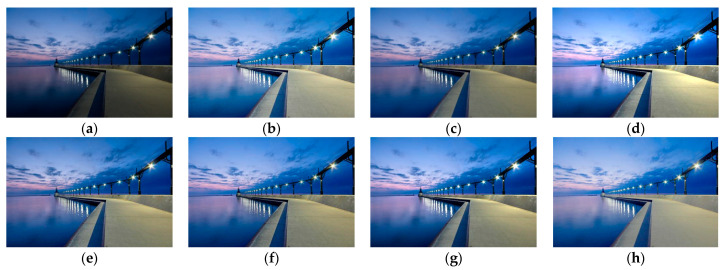
Comparing enhanced results of Venice with different methods. (**a**) Input image, (**b**) enhanced with LECARM, (**c**) enhanced with FFM, (**d**) enhanced with LIME, (**e**) enhanced with AFEM, (**f**) enhanced with JIEP, (**g**) enhanced with SDD, (**h**) enhanced with proposed RBFA method.

**Figure 16 entropy-23-00746-f016:**
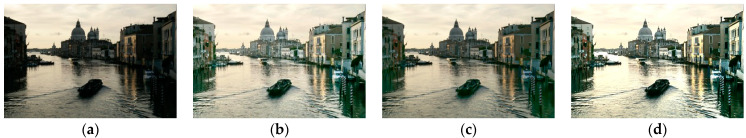
Comparing enhanced results of Venice with different methods. (**a**) Input image, (**b**) enhanced with LECARM, (**c**) Enhanced with FFM, (**d**) enhanced with LIME, (**e**) enhanced with AFEM, (**f**) enhanced with JIEP, (**g**) enhanced with SDD, (**h**) enhanced with proposed RBFA method.

**Table 1 entropy-23-00746-t001:** Time cost of different methods.

Image Size	100 × 100	700 × 700	1300 × 1300	1900 × 1900	2500 × 2500	3100 × 3100	3700 × 3700	4300 × 4300
LECARM	0.151	0.396	0.707	1.234	1.934	2.823	3.951	5.398
AFEM	0.048	0.204	0.566	1.136	2.014	3.075	4.674	5.959
LIME	0.030	0.124	0.394	0.825	1.437	2.203	3.209	4.363
FFM	0.182	5.043	17.071	36.819	65.744	95.577	142.183	197.190
SDD	0.222	8.882	34.930	79.808	139.754	209.301	345.587	526.162
JIEP	0.079	3.565	13.332	29.159	45.297	55.327	82.677	120.519
Proposed	**0.013**	**0.071**	**0.249**	**0.519**	**0.909**	**1.419**	**2.076**	**2.804**

**Table 2 entropy-23-00746-t002:** Results of image quality measure metrics with different methods.

Metrics	LECARM	AFEM	FFM	JIEP	LIME	SDD	Proposed
PIQE	39.818	39.809	42.884	40.072	42.705	51.457	**38.601**
LOE	415.594	253.646	291.906	296.568	749.862	493.806	**7.660**
MSE	3777.2175	2021.305	2823.849	2241.768	**1153.584**	1617.479	1158.174
SSIM	0.531	0.747	0.709	0.732	0.739	0.751	**0.753**
PSNR	12.504	16.350	14.464	15.847	18.136	17.511	**18.258**

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
