# Peer review of "Retinex-Based Fast Algorithm for Low-Light Image Enhancement"

_entropy, 2021, doi:10.3390/e23060746_

Round 1

Reviewer 1 Report

This paper proposed the Retinex-based fast enhancement method, which helps uneven brightness and greatly improves the brightness of low-light areas in figures.

Key  strengths:

  1. The paper is overall well written
  2. The authors have provided the algorithm, approach, and experiment in a detailed format
  3. The comparison with the state-of-the-art works have been provided

Author Response

Comments and Suggestions for Authors :

This paper proposed the Retinex-based fast enhancement method, which helps uneven brightness and greatly improves the brightness of low-light areas in figures.

Key strengths:

  1. The paper is overall well written
  2. The authors have provided the algorithm, approach, and experiment in a detailed format
  3. The comparison with the state-of-the-art works have been provided

Response: Thank you for your comments. We wish you a wonderful day!

Reviewer 2 Report

The authors present a well-written and well-organized paper. The scientific novelty of the proposed approach is sufficient. The solution looks to be light and elegant, works faster than some listed alternatives.

The charts in Figure 9 requires axis labels. it seems, that the vertical axis shows seconds, however, it seems the big "S" is used and it is unusual.horizontal axis has no label at all.

Author Response

Comments and Suggestions for Authors :

The authors present a well-written and well-organized paper. The scientific novelty of the proposed approach is sufficient. The solution looks to be light and elegant, works faster than some listed alternatives.

Point 1: The charts in Figure 9 requires axis labels. it seems, that the vertical axis shows seconds, however, it seems the big "S" is used and it is unusual. horizontal axis has no label at all.

Response: Thank you for your suggestion. We have given the axis labels in Figure 9, and changed the big "S" to "s".
